# Association of High-Risk Obstructive Sleep Apnea with Artificial Intelligence-Guided, CT-Based Severity Scores in Patients with COVID-19 Pneumonia

**DOI:** 10.3390/jcm13216415

**Published:** 2024-10-26

**Authors:** Zeynep Atceken, Yeliz Celik, Cetin Atasoy, Yüksel Peker

**Affiliations:** 1Department of Radiology, Koc University School of Medicine, Istanbul 34010, Türkiye; zatceken@kuh.ku.edu.tr (Z.A.); catasoy@kuh.ku.edu.tr (C.A.); 2Department of Pulmonary Medicine, Koc University School of Medicine, and Koc University Research Center for Translational Medicine (KUTTAM), Koc University, Istanbul 34010, Türkiye; yecelik@ku.edu.tr; 3Department of Molecular and Clinical Medicine, Sahlgrenska Academy, University of Gothenburg, 40530 Gothenburg, Sweden; 4Department of Clinical Sciences, Respiratory Medicine and Allergology, Faculty of Medicine, Lund University, 22185 Lund, Sweden; 5Division of Pulmonary, Allergy, and Critical Care Medicine, University of Pittsburgh School of Medicine, Pittsburgh, PA 15213, USA

**Keywords:** OSAS, COVID-19, chest CT, artificial intelligence

## Abstract

**Background:** We have previously demonstrated that high-risk obstructive sleep apnea (HR-OSA), based on a modified Berlin Questionnaire (mBQ), is linked to worse clinical outcomes. Chest computed tomography (CT) imaging with the implementation of an artificial intelligence (AI) analysis program has been a valuable tool for the speedy assessment of huge numbers of patients during the COVID-19 epidemic. In the current study, we addressed how the severity of AI-guided, CT-based total opacity ratio (TOR) scores are associated with high-risk OSA and short-term outcomes in the same cohort. **Methods:** The ratio of the volume of high opacity areas to that of the total lung volume constituted the TOR. We arbitrarily applied thresholds of <5 (no or mild TOR), ≥5 and <15 (moderate TOR), and ≥15 (severe TOR). **Results:** In total, 221 patients were included. HR-OSA was observed among 11.0% of the no or mild TOR group, 22.2% of the moderate TOR group, and 38.7% of the severe TOR group (*p* < 0.001). In a logistic regression analysis, HR-OSA was associated with a severe TOR with an adjusted odds ratio of 3.06 (95% confidence interval [CI] 1.27–7.44; *p* = 0.01). A moderate TOR predicted clinical worsening with an adjusted hazard ratio (HR) of 1.93 (95% CI 1.00–3.72; *p* = 0.05) and a severe TOR predicted worsening with an HR of 3.06 (95% CI 1.56–5.99; *p* = 0.001). **Conclusions:** Our results offer additional radiological proof of the relationship between HR-OSA and worse outcomes in patients with COVID-19 pneumonia. A TOR may also potentially indicate the individuals that are at higher risk of HR-OSA, enabling early intervention and management strategies. The clinical significance of TOR thresholds needs further evaluation in larger samples.

## 1. Introduction

Coronavirus disease 2019 (COVID-19) has caused a major global public health crisis in the last four years. According to the World Health Organisation, COVID-19 has affected more than 775 million people, with more than 7 million deaths (as of 26 May 2024) [1]. The most frequent comorbidities have been identified as hypertension, diabetes, and cardiovascular disease [2,3], as well as obesity [4,5].

Obstructive sleep apnea (OSA) is characterized by intermittent cessations of breathing during sleep, resulting in recurrent hypoxemia and arousals, and the condition is strongly associated with obesity, hypertension, and diabetes [6,7]. It has been proposed that patients with OSA may be predisposed to COVID-19 infection with poorer outcomes [8,9,10]. It has also been demonstrated that COVID-19 cases with known OSA diagnoses had higher mortality rates [11]. The real prevalence of OSA in patients with COVID-19 infection is uncertain given that objective sleep studies with polysomnography (PSG) are not realistic during an acute respiratory infection with a high transmission rate. In 2020, we conducted a questionnaire-based cohort study among 320 patients diagnosed with COVID-19 and anticipated the occurrence of high-risk OSA as 38% based on the Berlin Questionnaire (BQ) [12]. Since obesity and hypertension are comorbidities known to have unfavorable effects on the clinical course of COVID-19, we have also modified the BQ scoring system by excluding these conditions in order to better control for the prognostic role of high-risk OSA in patients with a history of COVID-19 infection. This modified BQ (mBQ) scoring system estimated the OSA prevalence as 22%. Thus, patients with modified high-risk OSA had worse clinical outcomes compared with those with low-risk OSA [12].

In a following validation study [13], the participants filled out the questionnaires again, and a PSG was conducted in the hospital. OSA was defined as an apnea–hypopnea index (AHI) of 15 events/h or more, and based on the PSG results, the mBQ had a sensitivity of 89%, a specificity of 93%, and an accuracy of 91% [13].

Machine learning, the basis of artificial intelligence (AI), is a branch of computer science that focuses on the development of human cognitive processes. AI uses algorithms that are programmed to learn, perform tasks, or solve particular problems automatically and can be used to create data-based models to aid in decision-making [14]. In recent decades, AI approaches have been effectively incorporated into imaging modalities frequently employed in the assessment of cardiovascular disorders, including computed tomography (CT), echocardiography, and cardiac magnetic resonance imaging [15].

Likewise, during the COVID-19 pandemic, chest CT imaging, combined with the use of an AI analysis algorithm, proved to be a useful technique for quickly evaluating a large number of patients. In a post-hoc study of the previously stated cohort, we recently evaluated the association between short-term outcomes and AI-guided, CT-based severity scores (SS) [16]. The main outcome of the study was the requirement for hospitalization and supplementary oxygen during a four-week follow-up period. The SS was defined as the ratio of the volume of high opacity areas to that of the entire lung volume (total opacity ratio [TOR]). The relationship between SS and the need for extra oxygen was analyzed using the receiver operating characteristics (ROC) curve, and the results showed a cut-off score of 2.65 on the CT images. The sensitivity was 81% and the specificity was 56%. In a multivariate logistic regression model, SSs >2.65 were related to an almost 4-fold risk increase for needing extra oxygen, and with a 2.4-fold risk increase for requiring hospitalization [16].

In the current study, we defined the TOR thresholds arbitrarily as no or mild TOR (<5), moderate TOR (≥5 and <15), and severe TOR (≥15). We evaluated whether the association of high-risk OSA with TOR severity could provide additional proof of the possible relationship between OSA and adverse outcomes in COVID-19 infections, and we further addressed the predictive value of TOR thresholds for clinical worsening.

## 2. Materials and Methods

### 2.1. Study Participants

This study was a double-center, post-hoc evaluation of a 2020 COVID-19 cohort. In all cases, COVID-19 diagnosis was based on real-time polymerase chain reaction (PCR) testing of samples collected via oropharyngeal or nasopharyngeal swabs from individuals with a suspected infection (e.g., exposure, symptoms, radiological imaging, and laboratory test findings). In the current study, 320 out of a total of 472 patients diagnosed with COVID-19 were included (Figure 1). The inclusion criteria were being an adult with a diagnosis of COVID-19 and having adequate CT scans for the AI analysis. Exclusion criteria were signs of tuberculosis on the CT with dense fibrotic scars and large atelectasis, a lack of scans for the AI analysis, and primary or secondary lung malignancies, as the total aerated lung parenchyma would decrease the opacities from pneumonic infiltration. As illustrated in Figure 1, patients with these diseases (n = 23), as well as those with non-adequate chest CTs (n = 76), were excluded from the analysis. Demographic data, comorbidities, clinical symptoms, drugs, laboratory and radiological findings, treatment, and outcomes were collected prospectively from electronic health records. Participants filled out the BQ at the outpatient clinic, in the hospital, or shortly after discharge from the hospital. The clinical, laboratory, and radiological follow-ups were scheduled and the polysomnographic, echocardiographic, and lung function evaluations were planned within a year of initial COVID-19 onset [12]. The BQ has been used in general populations [17] and clinical cohorts [18] as a screening tool for OSA [6]. The modified high-risk (mHR) OSA score excluded obesity and hypertension and applied the three subcategories from the questionnaire: Subcategory 1 included snoring patterns, Subcategory 2 included breathing pauses, and Subcategory 3 included tiredness or sleepiness in the morning and/or daytime. Patients were classified as having HR-OSA when they received a positive score in at least two subcategories. Demographics, concomitant diseases, and medications at baseline and during the follow-up were documented. A body mass index (BMI) of at least 30 kg/m^2^ was defined as obesity [19].

### 2.2. Chest CT Scan Protocol and Assessment

All participants underwent scanning in a supine position after a full inspiratory breath-hold using a 64-detector row CT scanner (Somatom^®^ Definition AS; Siemens Healthineers, Forchheim, Germany). Intravenous contrast medium was not administered.

Image analysis to assess pneumonia severity was conducted using an automated lung opacity analysis program, called “CT Pneumonia Analysis”, from Siemens Healthineers (Forchheim, Germany). The algorithm was designed to automatically identify and quantify high opacity areas, such as ground glass opacity and consolidation, which commonly exist in lung infections. The algorithm calculated the lobe volume, total lung volume, and areas with high opacity, including ground glass and consolidation. The severity of involvement, represented as the percentage of total opacity, was determined by the ratio of the volume of high opacity areas to that of the total lung volume. Figure 2a,b illustrates two examples of AI and CT images showing TORs of 1.1 and 17.2, respectively.

This AI model has been validated by Chamberlin et al. [20] for lung cancer screening with respect to lung nodules and coronary calcium. Agreement with an expert reader was found to be excellent (Cohen’s kappa of lung nodule detection: 0.846, intraclass correlation coefficient of coronary calcium volume: 0.904), with a sensitivity of 100% and 92.9% and a specificity of 70.8% and 96.0% for the presence of lung nodules and coronary calcifications, respectively. Focusing on the other end of the spectrum of patient cohorts, namely, patients with complex lung disease, such as interstitial lung disease, chronic obstructive pulmonary disease, bronchitis, edema, or pulmonary embolism, Abadia et al. [21] investigated the accuracy of lung nodule detection and localization, and on a patient level, the system showed a sensitivity of 89.4% and a specificity of 82.5%.

#### 2.2.1. Sample Size

In the first study, the recovery rate of patients with COVID-19 with high-risk OSA was anticipated to be approximately 80% within four weeks of hospitalization, compared to 90% among patients with low-risk OSA [12]. The sample size for hospitalized participants was considered as 196 for the primary outcome. A particular power analysis was not conducted for the current protocol.

#### 2.2.2. Statistical Analysis

We used IBM SPSS Statistics for Windows, version 26.0 (IBM Corp., Addison, TX, USA, 2020) for statistical analysis. For the normality of continuous variables, Shapiro–Wilk’s test was applied. For descriptive statistics, means and standard deviations were used for normally distributed variables, and medians (and interquartile ranges [IQRs]) were used for anon-normally distributed variables. We applied nonparametric statistical methods for values with skewed distributions, and the Kruskal–Wallis test to compare the three groups. Regarding the comparison of two normally distributed groups, a Student’s *t*-test was used. The chi-square test was applied for categorical variables and is shown as observation counts (and percentages). The relationship between HR-OSA and the TOR severity was tested by applying a logistic regression analysis adjusted for age, BMI, sex, and hypertension. Moreover, a time-dependent Cox model with a test for proportional hazards was used to anticipate the impact of HR-OSA on clinical worsening. Statistical significance was accepted when two-sided *p* < 0.05.

## 3. Results

The average age of the entire cohort was 52.6 ± 15.6 (range, 23–93) years, and the mean BMI was 27.7 ± 5.0 (range, 17.4–46.9) kg/m^2^.

As shown in Table 1, HR-OSA was observed in 11.0% of the no or mild TOR group, 22.2% of the moderate TOR group, and 38.7% of the severe TOR group (*p* < 0.001). Patients in the moderate and severe TOR groups were significantly older and had higher BMIs than those in the no or mild TOR group. There was a significant dose–response relationship between the occurrence of hypertension, as well as cardiac disease, and the severity of the TOR. The need for hospitalization, supplemental oxygen, and use of the ICU ward was significantly different across the TOR severity categories. The usage of antiviral drugs was significantly higher among patients with a severe TOR (Table 1).

As shown in Table 2, age, hypertension, and HR-OSA were significantly related with a severe TOR, whereas sex, BMI, and diabetes were not.

As shown in Table 3, hypertension and HR-OSA remained significant in the multivariate model adjusted for age, sex, and BMI.

As illustrated in Figure 3, a moderate TOR was predictive of clinical worsening, with an almost 2-fold risk, and a severe TOR was associated with a 3-fold risk increase.

## 4. Discussion

The main finding of this study was that HR-OSA was associated with a severe TOR, with a 3-fold risk increase when adjusted for age, sex, BMI, and hypertension. Moreover, there was a dose–response relationship between the severity of the TOR and clinical worsening.

To our best knowledge, this is the first study using an AI-based CT scan opacity scoring system as a radiological tool for evaluating the severity of COVID-19 pneumonia, applying arbitrary TOR thresholds (<5, ≥5 and <15, ≥15) that may be used to determine clinical outcomes in future studies.

We have previously conducted a questionnaire-based prospective observational cohort study on the clinical outcomes of COVID-19 patients categorized as high- or low-risk for OSA based on the mBQ [12]. Our analysis suggested that patients with HR-OSA had poorer clinical outcomes compared to those with low-risk OSA, independent of sex, age, and concomitant diseases. In our multivariate regression analyses, HR-OSA and male sex were predictors of delayed clinical improvement. We also found that HR-OSA was associated with clinical deterioration and a greater need for supplemental oxygen in the entire study population.

A similar study design was conducted by Arish et al. [22], who showed that, based on the BQ, OSA in COVID-19 patients is associated with the severity of the disease. They reported that individuals with HR-OSA not only experienced more severe symptoms of COVID-19 but also had longer hospital stays compared to those with low-risk OSA. Breville and colleagues [23] found that patients with HR-OSA were 14 times more likely to develop severe COVID-19-associated acute encephalopathy (AE), experience a longer duration of COVID-19-associated AE, and develop a long-term disability. They used the modified NoSAS (Neck, Obesity, Snoring, Age, Sex) score to demonstrate that more severe AE occurred in HR-OSA patients [23].

Iannella at al. [24] investigated 96 hospitalized patients with severe COVID-19 infections, applying the STOP-Bang questionnaire. Among participants, 42% of them requiring respiratory assistance were at a high risk of having OSA, compared to 21% of patients requiring conventional oxygen therapy. They divided the patients into two groups (Group 1 and Group 2) based on disease severity. A previous history of OSA was reported in 21% of Group 1 patients and 29% of Group 2 patients. The authors proposed that the likelihood of a severe case of COVID-19 increases in patients at a high risk of OSA [24]. These findings closely resemble those demonstrated by Bhatraju et al., where 21% of patients who were hospitalized for severe COVID-19 pneumonia had a history of OSA [25], and Arentz et al., who reported that 29% of critically/severely ill COVID-19 patients had a history of snoring and witnessed apnea [26].

During the COVID-19 pandemic, CT imaging with the use of an AI analysis program has proven to be a useful technique for the prompt assessment of a large number of patients. Deep learning methods were modified in the work by Harmon et al. [27] to identify COVID-19 pneumonia in chest CT scans. The AI algorithms in that study had an accuracy of 91%, sensitivity of 84%, and specificity of 93% in distinguishing COVID-19-related findings from other clinical entities. In another study, the diagnostic utility of AI had an accuracy of 92% [28]. In a study by Mei et al. [29], an AI system was trained on a dataset of more than 5000 cases, including confirmed COVID-19 patients and controls, and its accuracy rate was 96%.

An AI system’s diagnostic performance has also been compared to that of human readers [30], demonstrating that radiologists and AI software performed similarly in identifying SARS-CoV-2 pneumonia.

In a different study, a deep learning-based CT severity score’s sensitivity and specificity for identifying COVID-19 pneumonia were reported as 39.0% and 80.2%, respectively [31].

In another study, an automated MDCT pneumonia analysis program was used to assess the prognostic significance of 96 hospitalized patients with COVID-19 pneumonia as an early outcome predictor [32]. According to the authors, the program quickly determined the degree of COVID-19 pneumonia illness and had a strong correlation with both clinical outcomes and the severity of the illness.

Afshar-Oromieh et al. [33] stated that AI, especially machine learning, has emerged as an effective method that performs comparably to human practitioners in the differential diagnosis of COVID-19 pneumonia. Tasmi et al. [34] used a machine learning model to predict the mortality risk of individuals infected with COVID-19 who also had OSA. Through the analysis of patient medical histories to assess their future risk of death, such models could assist in prioritizing treatment for individuals with a higher risk, potentially reducing overall mortality rates among vulnerable patients with preexisting medical conditions. It has been suggested that CT severity scores and TORs calculated by AI showed good diagnostic accuracy in determining mortality and ventilation criteria [35]. In our previous study [16], we showed that the association between the TOR in CT images and the need for supplemental oxygen revealed a cut-off score of 2.65 with a sensitivity of 81% and a specificity of 56%.

AI-based calculations and severity scores may predict a need for supplemental oxygen and hospitalization in patients with pneumonia. Patients with high TOR values are likely to exhibit more severe symptoms and complications. This underscores the potential of AI in evaluating patients’ risk factors and aiding with treatment decisions. Studies indicate that AI-based severity scores, in addition to clinical parameters, can be a valuable tool in gaining a better understanding of the clinical course of COVID-19 pneumonia. These findings highlight the growing role and value of AI in medical diagnosis and management. This can be helpful in enhancing patient care, research methodologies, and the overall quality of healthcare delivery [36,37].

The findings of our study highlight the potential for AI-guided CT imaging to assess the severity of OSA and OSA’s impact on short-term outcomes in patients with COVID-19 pneumonia. Additionally, the integration of AI-based studies in routine clinical practice may enhance risk stratification and optimize patient management strategies, ultimately improving outcomes in this patient population.

Our study has some limitations. First, the study included 221 patients, which might limit the generalizability of findings. A larger sample size might provide more robust results and allow for better generalization to the broader population. Second, the study was conducted in the central area of Istanbul, which could affect the diversity of the patient population and the generalizability of the findings to other settings or urban populations. Third, while the study design was adjusted for age, sex, BMI, and hypertension, there might be other confounders that could influence the association between HR-OSA and TOR scores, as well as short-term outcomes. Fourth, the study focused on short-term outcomes, and thus, the long-term outcomes of TOR scores on COVID-19 pneumonia are not addressed.

## 5. Conclusions

Our results provide additional proof of the association between OSA and worse outcomes in patients with COVID-19 pneumonia with regard to radiological severity scores. The significant association of HR-OSA with a severe TOR may help to predict clinical deterioration. Furthermore, the TOR may also potentially indicate which individuals are at a higher risk of HR-OSA, enabling early intervention and management strategies. The integration of AI into CT imaging holds great promise for the future of risk assessment and prevention and has the potential to automate and rationalize the interpretation of imaging studies, allowing for faster and more accurate risk stratification. As reviewed in detail in a recent paper [15], AI technology can assist in the identification of subtle abnormalities in imaging that may not be apparent to the human eye, further improving risk assessment and patient care. Further research with larger sample sizes is needed to validate the clinical significance of the TOR thresholds identified in the current study.

## Figures and Tables

**Figure 1 jcm-13-06415-f001:**
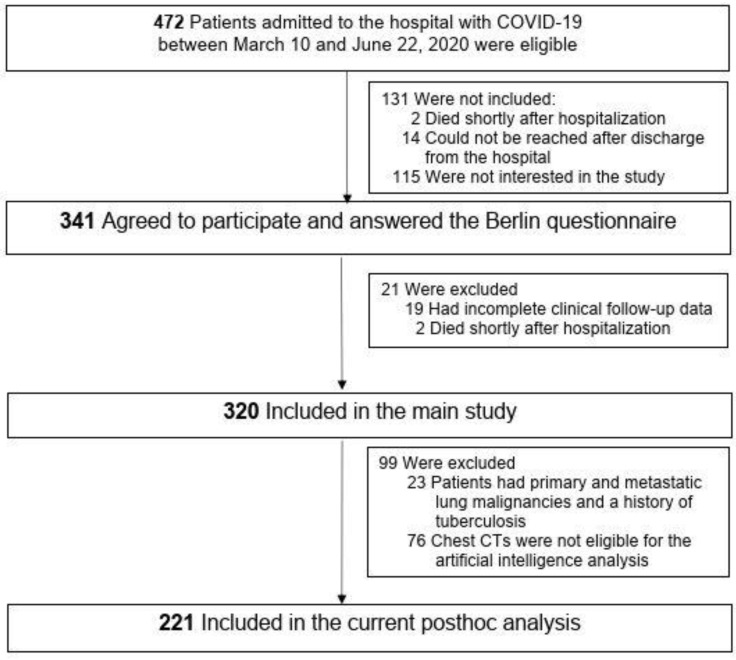
Flow-chart. COVID-19 = coronavirus disease; CT = computed tomography.

**Figure 2 jcm-13-06415-f002:**
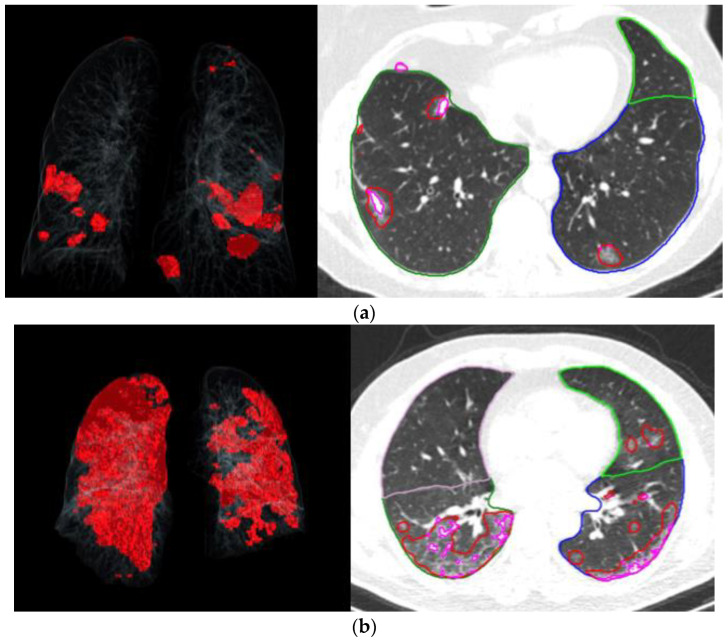
(**a**)—A volume rendering image (**left**) and a representative axial image (**right**) showing affected areas and depictions of segmented lung lobes and high opacity areas of a participant with a total opacity ratio of 1.1. (**b**)—A volume rendering image (**left**) and representative axial image (**right**) showing affected areas and depictions of segmented lung lobes and high opacity areas of a participant with a total opacity ratio of 17.2.

**Figure 3 jcm-13-06415-f003:**
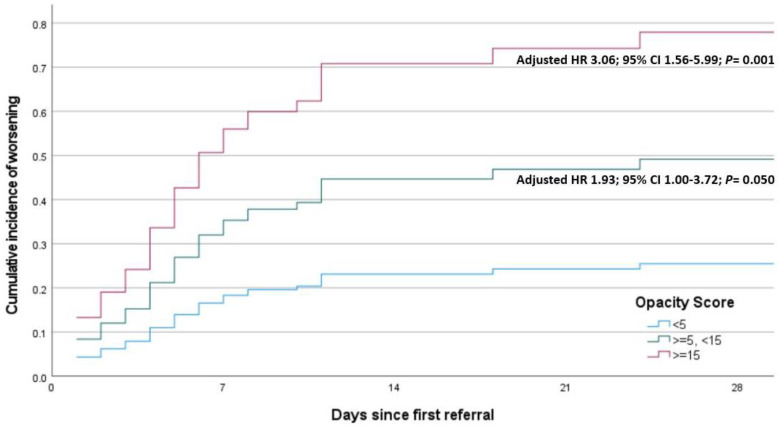
Cumulative incidence of clinical worsening in the study population of 221 cases with coronavirus disease (COVID-19), adjusted for covariates. CI = confidence interval; HR = hazard ratio; <5 = no or mild TOR; >=5, <15 = moderate TOR; >=15 = severe TOR.

**Table 1 jcm-13-06415-t001:** Demographic and clinical characteristics of the study participants.

Variables	<5 No or Minimal TOR N = 136	≥5, <15 Mild to Moderate TOR N = 54	≥15 Severe TOR N = 31	*p*-Value
HR-OSA	15 (11.0)	12 (22.2)	12 (38.7)	<0.001
Demographic Characteristics Age ≥ 65 yrs	15 (11.0)	20 (37.0)	8 (25.8)	<0.001
Age, yrs	48.1 (37.4–58.4)	58.7 (51.5–72.0)	56.4 (49.4–66.7)	<0.001
Sex	79 (58.1)	30 (55.6)	21 (67.7)	0.526
BMI, kg/m^2^	25.9 (24.0–29.9)	29.0 (25.8–32.9)	28.3 (25.8–31.2)	0.001
Comorbidities Current Smoking	15 (11.9)	4 (7.7)	0 (0.0)	0.123
Obesity	34 (25.0)	24 (44.4)	10 (32.3)	0.032
Hypertension	32 (23.5)	24 (44.4)	19 (61.3)	<0.001
Diabetes	15 (11.0)	14 (25.9)	3 (9.7)	0.022
CAD	5 (3.7)	8 (14.8)	4 (12.9)	0.017
CHF	1 (0.7)	0 (0.0)	2 (6.5)	0.028
Atrial Fibrillation	1 (0.7)	1 (1.9)	1 (3.2)	0.522
Cardiac Disease	6 (4.4)	9 (16.7)	7 (22.6)	0.002
COPD	3 (2.2)	1 (1.9)	1 (3.2)	0.917
Asthma	3 (2.2)	1 (1.9)	1 (3.2)	0.917
Lung Disease	6 (4.4)	2 (3.7)	2 (6.5)	0.837
Cerebrovascular Disease	0 (0.0)	0 (0.0)	3 (9.7)	<0.001
Dementia	1 (0.7)	2 (3.7)	0 (0.0)	0.219
Psychiatric Disease	2 (1.5)	1 (1.9)	0 (0.0)	0.764
Known OSA	0 (0.0)	1 (1.9)	1 (3.2)	0.162
Malignancy	6 (4.4)	3 (5.6)	0 (0.0)	0.436
Chronic Kidney Disease	2 (1.5)	3 (5.6)	2 (6.5)	0.185
Hyperlipidemia	2 (1.5)	6 (11.1)	2 (6.5)	0.013
Hypothyroidism	7 (5.1)	5 (9.3)	3 (9.7)	0.470
Hospitalization	80 (58.8)	43 (79.6)	29 (93.5)	<0.001
Supplemental Oxygen	17 (12.5)	17 (31.5)	24 (77.4)	<0.001
ICU	4 (2.9)	4 (7.4)	14 (6.3)	0.003
Drug Treatment Drug treatment for COVID	91 (91.2)	53 (98.1)	30 (96.8)	0.153
Statins	3 (2.2)	9 (16.7)	2 (6.5)	0.001
Immune Suppressive	3 (2.2)	0 (0.0)	0 (0.0)	0.387
Chloroquine	109 (80.1)	50 (92.6)	29 (93.5)	0.034
Azithromycin	57 (41.9)	36 (66.7)	17 (54.8)	0.007
Favipiravir	21 (15.4)	12 (22.2)	15 (48.4)	<0.001
Oseltamivir	30 (22.1)	12 (22.2)	15 (48.4)	0.008
Ritonavir/Lopinavir	3 (2.2)	2 (3.7)	5 (16.1)	0.003
Tocilizumab	9 (6.6)	11 (20.4)	10 (32.3)	<0.001
Systemic Steroids	1 (0.7)	2 (3.7)	0 (0.0)	0.219
Anticoagulant	54 (39.7)	34 (63.0)	19 (61.3)	0.005

Definition of abbreviations: BMI = body mass index; CAD = coronary artery disease; CHF = congestive heart failure; COPD = chronic obstructive pulmonary disease; COVID-19 = coronavirus disease; ICU = intensive care unit; OSA = obstructive sleep apnea.

**Table 2 jcm-13-06415-t002:** Variables associated with a severe TOR in an unadjusted logistic regression model.

Odds Ratio		95% CI for Odds Ratio		*p*-Value
		Lower	Upper	
Age	1.034	1.008	1.060	0.01
BMI	1.032	0.959	1.110	0.40
Male sex	1.561	0.697	3.494	0.28
Hypertension	3.789	1.724	8.324	<0.001
Diabetes mellitus	0.595	0.170	2.086	0.41
HR-OSA	3.813	1.663	8.740	0.002

Definition of abbreviations: BMI = body mass index; HR = high-risk; OSA = obstructive sleep apnea.

**Table 3 jcm-13-06415-t003:** Variables associated with a severe TOR in the multivariate logistic regression analysis.

Odds Ratio		95% CI for Odds Ratio		*p*-Value
		Lower	Upper	
Age	1.016	0.986	1.047	0.30
BMI	0.979	0.899	1.066	0.62
Male sex	1.464	0.626	3.428	0.38
Hypertension	2.926	1.216	7.043	0.017
HR-OSA	3.068	1.265	7.440	0.013

Definition of abbreviations: BMI = body mass index; HR = high-risk; OSA = obstructive sleep apnea.

## Data Availability

Individual participant data reported in this article can be provided by contacting the corresponding author at yuksel.peker@lungall.gu.se.

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
