# Peer review of "Association of High-Risk Obstructive Sleep Apnea with Artificial Intelligence-Guided, CT-Based Severity Scores in Patients with COVID-19 Pneumonia"

_jcm, 2024, doi:10.3390/jcm13216415_

Round 1

Reviewer 1 Report

Comments and Suggestions for Authors

1.The description of AI tools used for CT image analysis is adequate, but including specifics about the AI model’s validation, accuracy, and any limitations would strengthen the reliability of the results.
2.The discussion of logistic regression results could include a discussion of any potential confounders that were adjusted for and how they might affect the study's findings.

Comments on the Quality of English Language

Moderate editing of English language required

Author Response

Reviewer #1:

1.The description of AI tools used for CT image analysis is adequate, but including
specifics about the AI model’s validation, accuracy, and any limitations would
strengthen the reliability of the results.

Reply: These statements are now included in the methods section: ‘’This AI model has
been validated by Chamberlin et al. (20) for lung cancer screening with respect to lung
nodules and coronary calcium. The agreement with expert reader has been found
excellent (Cohen’s kappa of lung nodule detection: 0.846, intraclass correlation
coefficient of coronary calcium volume: 0.904) at a sensitivity of 100 % and 92.9 %
(presence of lung nodules and presence of coronary calcifications, respectively) and a
specificity of 70.8 % and 96.0 %, respectively. Focusing on the other end of the
spectrum of patient cohorts, namely patients with complex lung disease such as
interstitial lung disease, chronic obstructive pulmonary disease, bronchitis, edema, and
pulmonary embolism, Abadia et al.(21) investigated the accuracy of the lung nodule
detection and localization and on a patient level, the system showed a sensitivity of
89.4 % and a specificity of 82.5 %.’’

2.The discussion of logistic regression results could include a discussion of any
potential confounders that were adjusted for and how they might affect the study's
findings.

Reply: In the multivariate logistic regression analysis, we have adjusted our findings
for age, sex, BMI and hypertension as the potential confounders. Other comorbidities
including current smoking, diabetes, cardiac disease, current smoking were not
associated with severe TOR in univariate analysis (now added as a separate Table),
and were therefore not included in the multivariate model. 

Reviewer 2 Report

Comments and Suggestions for Authors

The study investigated the association of the severity of the AI guided CT-based total opacity ratio scores with high-risk obstructive sleep apnea and short-term outcomes in the same cohort. which is an important issue and valuable in the clinical medicine and respiratory tract infections and radiology.

Based on the definition of the total opacity ratio as the ratio of the volume of high opacity areas to that of the total lung volume, the study arbitrarily applied thresholds total opacity ratio values. The study results suggest an additional radiological proof of the association of HR-OSA with worse outcomes in patients with COVID-19 pneumonia.

Title is informative and reliable.

However, some points should be considered:

Abstract:

Delete the subtitle “Rationale”

Improvement of conclusion is recommended.

Introduction:

Is satisfied, can optionally in-depth enriched.

Material and methods:

The beginning of the sentence “320 out of total 472 patients diagnosed “should written in words not number. This should be applied in the whole manuscript.

Conclusion should also be improved; it should be the final word on the findings and hence presented in an appropriate style supported by the data.

Comments on the Quality of English Language

Needs minor changes.

Author Response

Reviewer #2:

The study investigated the association of the severity of the AI guided CT-based total opacity
ratio scores with high-risk obstructive sleep apnea and short-term outcomes in the same
cohort. which is an important issue and valuable in the clinical medicine and respiratory
tract infections and radiology.
Based on the definition of the total opacity ratio as the ratio of the volume of high opacity
areas to that of the total lung volume, the study arbitrarily applied thresholds total opacity
ratio values. The study results suggest an additional radiological proof of the association of
HR-OSA with worse outcomes in patients with COVID-19 pneumonia.
Title is informative and reliable.

Reply: We thank the reviewer for the positive comments.

However, some points should be considered:
Abstract:
Delete the subtitle “Rationale”

Reply: We replaced the “Rationale” with “Background”.

Improvement of conclusion is recommended.

Reply: We have added “TOR may also potentially indicate the individuals at higher risk of
HR-OSA, enabling early intervention and management strategies” in the conclusion.

Introduction:
Is satisfied, can optionally in-depth enriched.

Reply: We have added: ‘’Artificial intelligence (AI) is an arm of computer science with focus
on development of human cognitive processes, and machine learning is the basis of AI, which
uses databased models to facilitate decision-making and algorithms programmed to learn,
conduct tasks, or solve specific troubles automatically (ref 14). During the last decades, AI
techniques have been seamlessly integrated into imaging tools commonly used in the
evaluation of cardiovascular diseases such as computed tomography (CT), echocardiography,
and cardiac magnetic resonance imaging (ref 15). Similarly, chest CT imaging….’’ In the
Introduction session.

Material and methods:
The beginning of the sentence “320 out of total 472 patients diagnosed “should written in
words not number. This should be applied in the whole manuscript.

Reply: Done.

Conclusion should also be improved; it should be the final word on the findings and hence
presented in an appropriate style supported by the data.

Reply: We have added: ‘’Furthermore, TOR may also potentially indicate the individuals at
higher risk of HR-OSA, enabling early intervention and management strategies. The
integration of AI into CT-imaging holds great promise for the future of risk assessment,
prevention, and has the potential to automate and rationalize the interpretation of imaging
stud-ies, allowing for faster and more accurate risk stratification. As reviewed in detail in a
recent paper (15), the AI technology can assist in the identification of subtle abnormal-ities on
imaging studies that may not be apparent to the human eye, further improving risk assessment
and patient care.

Reviewer 3 Report

Comments and Suggestions for Authors

We thank the authors for submitting their manuscript focusing on the Association of High-Risk Obstructive Sleep Apnea with Artificial Intelligence Guided CT-based Severity Scores in Patients with COVID-19 Pneumonia. The results presented are intriguing and discussed thoroughly. However, minor revisions are necessary for clarity and robustness of the analysis:

1. We recommend the inclusion of a graphical abstract to concisely summarize the key findings and messages of the article, enhancing the readers' understanding at a glance.

2. In Table 1, we suggest specifying the sex variable as "male" or "female." A differentiated analysis by sex should be performed, highlighting any potential gender differences concerning the total opacity ratio within the studied population.

3. The variable "Cardiac disease" in Table 1 appears overly generalized. We advise its removal or a more precise definition to provide clearer insights.

4. The introduction should mention the employment of artificial intelligence not only for chest CT but also for cardiac CT. We encourage the authors to reference the work by Trimarchi et al. (2024) (PMCID: PMC11355021), which discusses the role of non-invasive imaging in cardiovascular prevention, as it aligns well with the manuscript's theme.

These enhancements would significantly contribute to the manuscript's overall quality and persuasiveness. We look forward to seeing the revised version.

Author Response

Reviewer #3:

We thank the authors for submitting their manuscript focusing on the Association of HighRisk Obstructive Sleep Apnea with Artificial Intelligence Guided CT-based Severity Scores in
Patients with COVID-19 Pneumonia. The results presented are intriguing and discussed
thoroughly. However, minor revisions are necessary for clarity and robustness of the
analysis:

Reply: We thank the reviewer for the positive comments.

1. We recommend the inclusion of a graphical abstract to concisely summarize the key
findings and messages of the article, enhancing the readers' understanding at a glance.

Reply: We have now included a graphical abstract as a supplement.

2. In Table 1, we suggest specifying the sex variable as "male" or "female." A differentiated
analysis by sex should be performed, highlighting any potential gender differences
concerning the total opacity ratio within the studied population.

Reply: It was a typo; it was male sex. Now specified in the table.

3. The variable "Cardiac disease" in Table 1 appears overly generalized. We advise its
removal or a more precise definition to provide clearer insights.

Reply: We agree that it was unnecessary so we have removed the ‘’cardiac disease’’ from the
table.

4. The introduction should mention the employment of artificial intelligence not only for chest
CT but also for cardiac CT. We encourage the authors to reference the work by Trimarchi et
al. (2024) (PMCID: PMC11355021), which discusses the role of non-invasive imaging in
cardiovascular prevention, as it aligns well with the manuscript's theme.

Reply: We thank the reviewer for this suggestion. We have now included the following
statements in the Introduction: ‘’Artificial intelligence (AI) is an arm of computer science
with focus on development of human cognitive processes, and machine learning is the basis
of AI, which uses databased models to facilitate decision-making and algorithms programmed
to learn, conduct tasks, or solve specific troubles automatically (14). During the last decades,
AI techniques have been seamlessly integrated into imaging tools commonly used in the
evaluation of cardiovascular diseases such as computed tomography (CT), echocardiography,
and cardiac magnetic resonance imaging (15). Similarly, …..’’

These enhancements would significantly contribute to the manuscript's overall quality and
persuasiveness. We look forward to seeing the revised version.

Reply: Thanks again.

Round 2

Reviewer 1 Report

Comments and Suggestions for Authors

No comment

Comments on the Quality of English Language

No comment

Author Response

Replied previously.